# Impact of *Salmonella enteritidis* Infection and Mechanical Stress on Antimicrobial Peptide Expression in *Hermetia illucens*

**DOI:** 10.3390/insects16070692

**Published:** 2025-07-04

**Authors:** Davide Santori, Anna Maria Fausto, Alessio Gelli, Anna Rita Pifferi, Samuele Dottarelli, Sofia Cucci, Francesca Di Donato, Goffredo Grifoni, Erminia Sezzi

**Affiliations:** 1Istituto Zooprofilattico Sperimentale del Lazio e della Toscana “M. Aleandri”, 00149 Roma, RM, Italy; alessio.gelli@izslt.it (A.G.); annarita.pifferi@izslt.it (A.R.P.); samuele.dottarelli@izslt.it (S.D.); sofia.cucci-esterno@izslt.it (S.C.); francesca.didonato@izslt.it (F.D.D.); goffredo.grifoni@izslt.it (G.G.); erminia.sezzi@izslt.it (E.S.); 2Department for Innovation in Biological, Agro-Food and Forest Systems (DIBAF), University of Tuscia, Via San Camillo De Lellis snc, 01100 Viterbo, VT, Italy; fausto@unitus.it

**Keywords:** *Hermetia illucens*, antimicrobial peptides, AMPs, BSF, *Salmonella enteritidis*, *Salmonella typhimurium*

## Abstract

Research on new molecules that can reduce the phenomenon of antibiotic resistance is one of the challenges of this century. Insects naturally produce antimicrobial peptides that can be used against bacteria, fungi, and viruses. *Hermetia illucens* (*Diptera*, *Stratiomyidae*) has distinguished itself by its ability to survive in substrates highly contaminated by microorganisms. Our work focuses on the production and action of antimicrobial peptides after infection of larvae with *Salmonella*. The results showed increased action compared to the control and mechanically stressed group.

## 1. Introduction

The use of antibiotics in livestock farming has been at the center of heated discussion for years due to its implications for public health [1], animal welfare [2], and the environment [3]. While these drugs are essential to treat infections in animals, their excessive and often indiscriminate use has contributed to one of the most serious health problems of our time: antibiotic resistance [4,5,6,7].

For decades, antibiotics have been used not only for therapeutic purposes but also as growth promoters [8], a practice that allows animals to gain weight more quickly and prevents disease under intensive farming conditions [9].

The European Union has adopted a very strict policy on the use of antibiotics in animal feed to combat antibiotic resistance and ensure food safety. In 2006, Regulation (EC) 1831/2003 [10] banned the use of antibiotics as growth promoters in livestock, thus preventing them from being used to improve the production performance of animals without a real therapeutic need [11]. With the entry into force of Regulation (EU) 2019/6 in 2022 [12], the use of these drugs has been further restricted, allowing them to be used only for well-justified veterinary purposes [13].

In addition, to protect human health, the EU has stipulated that antibiotics considered critical may not be used in livestock, to prevent their use in animals from compromising the effectiveness of these drugs in human medical treatment [14]. To ensure compliance with these rules, a monitoring and control system has been implemented that obliges farmers and veterinarians to register and justify the use of antibiotics.

*Salmonella* is one of the most important challenges to food safety and public health [15]; this is a particularly insidious bacterium because it can contaminate food of animal origin, especially meat and eggs, and cause serious human infections [16,17,18]. If strains of this pathogen develop resistance to commonly used antibiotics, treating infections becomes much more difficult, with potentially fatal consequences for patients with increased health vulnerability [14].

*Hermetia illucens*, commonly known as the black soldier fly, is emerging as a promising new source of antimicrobial peptides (AMPs) [5,19]. The larvae of this species are rich in antimicrobial peptides and other bioactive substances that have been shown to effectively combat a wide range of pathogenic bacteria, including those resistant to traditional antibiotics [20,21,22,23].

These peptides are part of the insect’s innate immune system [24] and have demonstrated significant activity against a wide range of pathogenic microorganisms [25,26,27]. Among the most studied, defensins have proven particularly effective due to their ability to perforate bacterial cell membranes, leading to the destruction of the pathogen [28,29,30,31].

In addition to defensins, other antimicrobial peptides produced by this species have shown strong application potential. Cecropins, for instance, act by altering the permeability of the bacterial membrane, with lethal effects on both Gram-negative and Gram-positive bacteria [28].

The particularity of these peptides is that their expression can vary in response to infection, a sign of an adaptive mechanism. Studies have shown that exposure of *H. illucens* to specific pathogens, including fungi and bacteria, can induce increased production of AMPs, making the insect an interesting model for understanding natural immune responses [32,33,34,35,36,37].

The aim of this work was to test whether a direct infection with *Salmonella enteritidis* or the exposure of larvae to a substrate contaminated with the same pathogen could lead to increased expression of AMPs in *Hermetia illucens* larvae. An experimental group of larvae was mechanically stressed to test whether this type of treatment could also lead to changes in expression.

The functionality and expression of AMPs from the peptide fraction of the hemolymph of *H. illucens* larvae were tested by means of antibiogram, minimum inhibitory concentration (MIC), and molecular expression tests.

## 2. Materials and Methods

### 2.1. Hermetia illucens Rearing

*H. illucens* larvae (HiL) were supplied by BEF biosystem s.r.l. (Casalnoceto, Italy) and were fed with a standard Gainesville diet (50% wheat bran, 30% alfalfa meal, and 20% corn) and grown under controlled humidity (70 ± 5%) and temperature (27 ± 1.0 °C) conditions.

### 2.2. Experimental Setup

Four experimental groups, each consisting of three replicates, were set up to test the effect of the AMPs produced by *H. illucens*.

Each experimental group consisted of a total of 600 HiL, divided into three replicates of 200 larvae each (Figure 1).

The first experimental group, called the pierced group (P), consisted of *H. illucens* larvae that were first washed with sterile water and subsequently stimulated by a puncture with a needle (0.5 mm diameter). In the second group, called the direct infection group (D), the larvae were washed with sterile water and infected by puncture with a needle (0.5 mm diameter) dipped in a suspension of *S. enteritidis* at a 4.0 McFarland concentration (12.0 × 10^8^ CFU/mL). The third group, named the indirect infection group (I), included larvae washed with sterile water and reared on a substrate contaminated with 50 mL of a suspension of *S. enteritidis* (ATCC-13076) at a 4.0 McFarland concentration (12.0 × 10^8^ CFU/mL). In the fourth group used as the control group (C), the larvae were not subjected to any treatment but were washed with sterile water

The experimental design included 3 sampling times: T0, when the larvae had not been treated, T1, 4 h after infection, and T2, 24 h after infection (Figure 2).

Larvae sampling was performed by collecting 60 HiL from each box for hemolymph extraction at each sampling time (T0, T1, and T2), for a total of 180 larvae for each experimental group. From the same boxes, at each sampling time, 3 larvae were also sampled for RNA extraction, with a total of 9 larvae for each experimental group.

### 2.3. Salmonella enteritidis and Salmonella typhimurium Strains

*Salmonella enteritidis* (ATCC-13076) and *Salmonella typhimurium* (ATCC-14028) strains were reconstituted from a cryobank system following standard microbiological protocols. The vial containing the strains was retrieved from storage at −80 °C and kept on ice to prevent premature thawing. Inside a biosafety cabinet, one cryobank bead was carefully removed using sterile forceps and transferred into a sterile test tube containing 10 mL of pre-warmed Tryptic Soy Broth (TSB) at 37 °C.

The test tube was then incubated at 37 °C for 24 h under static conditions to allow the bacteria to recover and grow. Following the incubation period, a small aliquot of culture was streaked onto Tryptic Soy Agar (TSA) plates using a sterile inoculation loop. The plates were subsequently incubated at 37 °C for an additional 24 h to promote the growth of distinct colonies.

After incubation, the plates were examined, and colonies consistent with the morphology of *Salmonella enteritidis*, smooth, round, and translucent, were observed. To confirm the purity and identity of the strains, a single well-isolated colony was selected and subjected to biochemical testing, including Triple Sugar Iron (TSI) agar and urea hydrolysis assays.

### 2.4. Hemolymph Collection

According to Scieuzo et al. [33], larval hemolymph was collected with a pipette (Rainin Instrument Co., Woburn, MA, USA) in ice-cold microtubes containing 0.015 g of l-ascorbic acid (Merck KGaA, Darmstadt, Germany) to prevent hemolymph melanization. The extracted hemolymph was centrifuged at 10,000 rcf for 5 min at 4 °C to recover only plasma and remove cellular components. The collected supernatant (cell-free hemolymph) was stored at −80 °C until use.

### 2.5. Peptide Isolation and Quantification

A precipitation protocol was used to isolate AMPs from the collected hemolymph using methanol (Merck Millipore, Burlington, MA, USA), acetic acid (Merck Millipore, Burlington, MA, USA), and water in a 90:1:9 *v*/*v* ratio [34].

This protocol allowed for the separation of putative AMPs from the higher molecular weight protein.

The sample was centrifuged at 16,000 rcf for 45 min at 4 °C. The supernatant obtained was then dried to remove organic solvents and resuspended in a volume of sterile water equal to the original sample volume.

In order to quantify the proteins present in the samples, a Bradford protein assay (Quick start kit Bio-Rad, Hercules, CA, USA) was carried out following the manufacturer’s instructions, absorbance at 595 nm of samples was measured using a spectrophotometer (Multiskan FC Microplate Photometer, Thermo Scientific, Milan, Italy).

All samples were stored at 4 °C +/− 2 °C.

### 2.6. Determination of the Minimum Inhibitory Concentration (MIC) of Hemolymph Against Salmonella enteritidis and Salmonella typhimurium

The minimum inhibitory concentration (MIC) test was performed against both *S. enteritidis* and *S. typhimurium*, testing all experimental conditions.

A standard solution of the peptide fractions was prepared at an initial concentration of 1.008 µg/µL. Serial twofold dilutions were performed in a 96-well plate using Mueller–Hinton broth, covering a range of concentrations from 1.008 μg/μL down to the lowest tested concentration 0.007 μg/μL.

A bacterial suspension was prepared from an overnight culture and adjusted to 0.5 McFarland standard (corresponding to approximately 1.5 × 10^8^ CFUs/mL) using a spectrophotometer at 600 nm (DeNovix DS–11, DeNovix Inc., Wilmington, NC, USA).

To each well containing the peptide fraction, 1 µL of the bacterial suspension was added.

Three controls were included in a 96-well plate: one containing the bacterial suspension only, the second containing Mueller–Hinton broth (MH-B) only, and the third containing the peptide fraction only.

The plate was incubated at 37 °C for 24 h. Following incubation, the MIC was defined using a spectrophotometer at a wavelength of 600 nm (Multiskan FC Microplate Photometer, Thermo Scientific, Milan, Italy).

### 2.7. Evaluation of the Antibacterial Activity of Hemolymph via an Antibiogram Assay

In vitro evaluation of the antimicrobial activity of the peptide fraction of the hemolymph was performed using an antibiogram test.

0.5 Mc Farland solutions of *S. enteritidis* and *S. typhimurium* were prepared and then evenly distributed onto plates containing MH-Agar, using a cotton swab. 5 μL of the peptide fractions of the hemolymph extracted from the experimental groups of larvae were dispensed onto the plates. As a negative control, 5 μL of sterile water was used. All tests were performed in triplicate, with the plate being incubated overnight at 37 °C.

### 2.8. Molecular Evaluation of Defensin and Cecropin Expression

The extraction of RNA from whole larvae was achieved through a process involving rapid freezing in liquid nitrogen, followed by grinding. This was followed by a sequence of washing and treatment with 75% isopropanol and ethanol. The resultant pellet was then resuspended in DEPC water.

Retrotranscription of RNA to cDNA was carried out using a commercial kit (High-Capacity cDNA Reverse Transcription Kit, Applied Biosystems, Waltham, MA, USA) following the manufacturer’s instructions. Quantification and qualitative evaluation of the retrotranscript was performed using a spectrophotometer (DeNovix DS–11, DeNovix Inc., Wilmington, NC, USA).

The relative expression of defensins and cecropins was evaluated using the 16 s gene as a reference gene. The ΔΔct was calculated using the control group as the base expression quantity.

The primer sequences used to verify the expression of defensin and cecropin are shown in Table 1.

### 2.9. Statistical Analysis

Prior to statistical comparisons, the assumptions of normality and homogeneity of variances were assessed. The normality of the data distribution within each group was tested using the Shapiro–Wilk test, while homogeneity of variances was evaluated using Levene’s test. When both assumptions were met, data were analyzed using one-way analysis of variance (ANOVA) to detect statistically significant differences between groups. In the case of significant ANOVA results, Tukey’s post hoc test was applied to adjust for multiple comparisons and control for type I errors. Specifically, in the case of minimum inhibitory concentration (MIC) assays, comparisons were conducted between each treatment group and the control group (bacterial strain), rather than among all groups simultaneously. All statistical analyses were performed using Python (version 3.1), and the results are presented as the means ± standard deviations (SDs). For MIC assays, a significance threshold of *p* < 0.001 was adopted; for relative expression analysis of cecropins and defensins, statistical significance was set at *p* < 0.005.

## 3. Results

### 3.1. Results of Minimum Inhibitory Concentration (MIC)

The hemolymph of the larvae at T0 was tested in order to verify its potential action (Figure 3).

At T0 for both *S. typhimurium* and *S. enteritidis*, the inhibitory concentration of the hemolymph protein fraction is 0.504 µg/µL.

The results obtained from the MIC tests against *S. enteritidis* at T1 and T2 are shown in Figure 4 and Figure 5.

At T1, the inhibitory concentration for *S. enteritidis* was 0.504 µg/µL for group C, 0.252 µg/µL for group P, 0.504 µg/µL for group I, and 0.126 µg/µL for group D.

At T2, the inhibitory concentration for *S. enteritidis* for group C was 0.504 µg/µL, for group (P), it was 0.126 µg/µL, for group (I), it was 0.252 µg/µL, and for group (D), it was 0.126 µg/µL.

The results obtained from the MIC tests against *S. typhimurium* at T1 and T2 are shown in Figure 6 and Figure 7.

At T1, the inhibitory concentration against *S. typhimurium* was 0.504 for group C, 0.504 for group (P), 0.504 for group (I), and 0.252 for group (D).

At T2, the inhibitory concentration against *S. typhimurium* was 0.504 for group C, 0.252 for group (P), 0.504 for group (I), and 0.126 for group (D).

### 3.2. Results of the Antibiogram Assay

The peptide fractions of hemolymph were tested to evaluate their antibacterial effect against *S. enteritidis* and *S. typhimurium* (Figure 8 and Figure 9) (Table 2 and Table 3).

The antibiogram test included a preliminary test (T0) performed on untreated larvae (LWT) at the beginning of the study, which showed the absence of inhibitory halos in the peptide fraction of the hemolymph against both *S. enteritidis* and *S. typhimurium*.

Antibiogram tests against *S. enteritidis* produced the following results: the hemolymph obtained from the control larvae (C) showed no inhibition halo at either T1 or T2 (Figure 8a,c).

Hemolymph from pierced larvae (P) produced halos of 4.1 mm, 3.5 mm, and 3.2 mm at T1; the presence of colonies within the halos was low. At T2, the halos were 2.1 mm, 1.7 mm, and 2.5 mm with a higher presence of colonies than at T1 (Figure 8a,c).

The hemolymph results for larvae placed in contact with the infected substrate (I) showed inhibition halos at T1 of 2.1 mm, 2.0 mm, and 1.9 mm, with significant presence of colonies inside; at T2, no halos were produced (Figure 8b,d).

The halos produced by the hemolymph extracted from the directly infected larvae (D) were 3.5 mm, 4.0 mm, and 3.2 mm with low colony presence in D1 and D2 and high presence in D3 at T1. At T2. the halos were 4.0 mm, 3.6 mm, and 3.4 mm in size, with low presence of colonies inside (Figure 8b,d).

Antibiogram tests against *S. typhimurium* produced the following results: the hemolymph obtained from the control larvae (C) showed no inhibition halo at either T1 or T2. (Figure 9a,c).

The hemolymph of the pierced larvae (P) at T1 produced halos of 4.0 mm, 4.1 mm, and 3.6 mm; the presence of colonies in the halos was variable, with lower presence in P1 and higher presence in P2 and P3. At T2, the halos were 2.1 mm, 2.0 mm, and 2.7 mm with a higher presence of colonies within halos (Figure 9a,c).

The hemolymph results for larvae placed in contact with the infected substrate (I) showed inhibition halos at T1 of 3.4 mm, 3.2 mm, and 3.1 mm, with significant presence of colonies inside; at T2, no halos were produced (Figure 9b,d).

The halos produced by the hemolymph extracted from the directly infected larvae (D) at T1 were 4.1 mm, 4.3 mm, and 3.5 mm in size, with low colony presence in D1 and D2 and high presence in D3. At T2, the halos were 2.5 mm, 2.2 mm, and 2.7 mm in size, with a high presence of colonies inside (Figure 9b,d).

### 3.3. Results of the Expression of Cecropins and Defensins

Analysis of the expression of cecropins under the three experimental conditions between T1 and T2 showed distinct trends. In the P group, relative expression remained constant over time. In contrast, the presence of *Salmonella* appeared to stimulate cecropin production, as observed in the I condition, where relative expression values increase from 1.3 to 3.0. This effect is even more pronounced in the D group, where relative expression increases from 2.0 to 5.3.

The results are shown in Figure 10 and Figure 11.

## 4. Discussion

In this work, we tested antimicrobial peptides derived from the protein fraction of hemolymph against two *Salmonella* strains. *Salmonella typhimurium* and *Salmonella enteritidis* are two of the most common serotypes responsible for foodborne infections in humans. These bacteria are among the main causes of salmonellosis, a food-borne illness that can cause severe gastrointestinal symptoms, especially in vulnerable individuals such as children, the elderly, and immunocompromised persons [16].

This choice of target organisms is consistent with previous studies highlighting the relevance of these serotypes in antimicrobial resistance surveillance programs [38].

The results obtained from the analysis of the antimicrobial activity of the larval hemolymph protein fraction highlight a consistent inhibitory effect against both *S. enteritidis* and *S. typhimurium*, with significant differences depending on the treatment applied and the time of sampling. Before the experimental treatments (T0), both bacterial strains exhibited the same minimum inhibitory concentration (MIC), indicating that the hemolymph possesses intrinsic antimicrobial activity, likely attributable to the constitutive presence of antimicrobial peptides or other defense-related proteins. As shown by Saadoun et al. [39], the hemolymph of insects has an underestimated amount of antimicrobial compounds present in their body components, such as lauric acid and chitin, and circulating metabolites in the form of AMP.

The intrinsic antimicrobial activity of hemolymph was also verified at the proteomic level by Makarova et al. [40] in *Tenebrio Molitor*.

A similar baseline antimicrobial effect has been reported in studies on insect immune systems, particularly that by Bulet et al. [41], who observed constitutive expression of AMPs in unstimulated larvae of various insects.

At T1, variations in MIC values against *S. enteritidis* became evident among the experimental groups. Specifically, the treated groups (Pierced larvae (P), larvae reared on a substrate contaminated (I), and larvae infected by puncture (D)) exhibited enhanced antimicrobial activity compared to the control, with MIC values progressively decreasing, reaching as low as 0.126 µg/µL in group D. This trend suggests activation of the larval innate immune system in response to specific stress, leading to the increased production or release of bioactive molecules with antibacterial properties. As shown by Vallet-Gely et al. [27], an insect’s immune system increases the production of antimicrobial molecules when it comes into contact with a pathogen. This increase is due to the activation of specific pathways, as shown by Ali Mohammadie Kojour et al. [42].

Comparable immune stimulation leading to enhanced antimicrobial output was described by Zhang et al. [43], who demonstrated that stress conditions can trigger the upregulation of AMP genes in insects.

A similar pattern was confirmed at T2, where MIC values continued to decrease in some groups, particularly P and D, indicating a markedly stronger antimicrobial potency compared to the control. The progressive reduction in the MIC over time implies either an adaptive-like immune response or, more plausibly, a reinforcement of innate immunity, possibly via mechanisms of immune priming.

This phenomenon mirrors observations in work by Little et al. [44], who proposed that immune priming in invertebrates leads to more efficient secondary immune responses upon repeated pathogen exposure.

Regarding *S. typhimurium*, the data at T1 showed a less pronounced effect of the treatments compared to *S. enteritidis*, with a noticeable MIC reduction only in group D. However, at T2, a more distinct response was observed: MIC values decreased in groups P and D, confirming that the treatments were also effective in enhancing hemolymph antimicrobial activity against this bacterial species. A comparison between the two *Salmonella* strains reveals that the response to treatment was more rapid and pronounced against *S. enteritidis*, while for *S. typhimurium*, the inhibitory effect became more evident over time.

The decrease in MIC values observed in our study is compatible with immune priming effects, already described in *H. illucens* by Nakagawa et al. [45], which showed an increase in antimicrobial peptide production after repeated exposure to bacterial stress.

Our results are in line with those of Auza et al. [46] who observed significant antimicrobial activity of peptides extracted from the insect *H. illucens* against the *S. typhimurium*. However, the authors of other studies, such as that by Romoli et al. [47], have reported variability in sensitivity between different *Salmonella* serotypes, suggesting that the structure of the bacterial outer membrane may influence the efficacy of peptides.

Overall, the data indicate that larval hemolymph is capable of modulating its antimicrobial activity in response to external stress [27,48], probably through the induction of effector molecules, and that this response is influenced by both the nature of the treatment and the duration of exposure [49]. The consistently stronger activity observed in group D at both time points suggests that this specific treatment plays a key role in enhancing larval innate immune defenses.

The results of the antibiogram assay against *S. enteritidis* revealed distinct trends in antimicrobial activity.

At T1, the control group did not display any visible inhibition of bacterial growth, confirming the absence of antimicrobial activity. In contrast, the condition involving group P exhibited clear signs of bacterial inhibition, with discernible halos forming around the application area. However, bacterial colonies were still present within the inhibited zones, suggesting that while the treatment exerted a certain degree of antimicrobial activity, it was not sufficient to completely prevent bacterial growth. Group I showed only limited inhibition, with much smaller halos and a pronounced presence of bacterial colonies even within these areas, indicating a relatively weak antimicrobial response. The D group, on the other hand, demonstrated inhibition similar in extent to the P group, but with noticeably fewer bacterial colonies, suggesting more effective suppression of bacterial growth in this condition.

At T2, a clear shift in the pattern of inhibition was observed. As expected, the control group continued to show no signs of antimicrobial activity. The P group, which initially exhibited moderate inhibition, now displayed reduced halo formation and a more pronounced presence of bacterial colonies, indicating that its antimicrobial effect had weakened over time, allowing bacterial growth [50]. In the I group, any previously observed inhibition had completely disappeared, with widespread bacterial growth confirming the transient and unsustained nature of its antimicrobial effect. In contrast, the D group maintained a more stable inhibitory response. Although there was a slight reduction in the extent of inhibition and a modest increase in bacterial presence, this condition continued to exert a measurable antimicrobial effect, especially when compared to the diminishing responses observed in the other experimental groups.

Taken together, these results suggest that the antimicrobial effect induced in the D group is not only more pronounced initially but also more persistent over time [51]. While P treatment provides an early inhibitory effect, its impact diminishes relatively quickly, allowing for bacterial regrowth. The I condition appears to be the least effective, with only limited inhibition at the beginning and no detectable effect later on. As anticipated, the control group remained entirely ineffective throughout the study.

A comparable trend was observed in the assays conducted against *Salmonella typhimurium*. At T1, the control group again showed no inhibition, consistent with the absence of antimicrobial activity. The P group showed the presence of moderate inhibition halos, though bacterial colonies were still evident within these areas, indicating incomplete suppression of growth. The I group exhibited a similar pattern, with some inhibition accompanied by persistent bacterial presence. Notably, the D group demonstrated the most effective response, with broader inhibition zones and comparatively reduced bacterial survival, pointing to a more robust antimicrobial action.

These findings are consistent with previous observations by Mastore et al. [52], who reported that the production and release of antimicrobial peptides can vary significantly depending on the method of immune stimulation

A comparable trend was observed in the assays conducted against *Salmonella typhimurium*. At T1, the control group again showed no inhibition, consistent with the absence of antimicrobial activity. The P group showed a presence of moderate inhibition halos, though bacterial colonies were still evident within these areas, indicating incomplete suppression of growth. The I group exhibited a similar pattern, with some inhibition accompanied by persistent bacterial presence. Notably, the D group demonstrated the most effective response, with broader inhibition zones and comparatively reduced bacterial survival, pointing to a more robust antimicrobial action.

These findings are consistent with previous observations by Mastore et al. [53], who reported that the production and release of antimicrobial peptides can vary significantly depending on the method of immune stimulation.

At T2, the persistence of antimicrobial activity varied notably among the treatment groups. The control group remained inactive, while the P condition showed a clear reduction in inhibition, along with increased bacterial growth. The I group lost all signs of antimicrobial activity, allowing for unimpeded bacterial proliferation. Although the D group also exhibited a decrease in inhibitory effect compared to the initial observation, the reduction was less pronounced, and bacterial growth remained relatively contained, indicating a more sustained antimicrobial response.

Overall, these results highlight a progressive decline in antimicrobial efficacy over time across all conditions [51]. Nevertheless, the D group consistently demonstrated the most prolonged and effective inhibition of bacterial growth. P treatment, while initially effective, failed to sustain its activity, providing only a transient and limited response. These findings underscore the importance of treatment modality in shaping the strength and duration of antimicrobial peptide activity [52].

These findings align with those reported by Hancock and Sahl [53], who emphasized that while AMPs are potent initially, their long-term effectiveness is often challenged by environmental degradation and bacterial adaptation strategies.

The complete disappearance of inhibition in the I condition suggests that it provides only short-term antimicrobial action. The D group, despite some decline, remains the most effective in suppressing *S. typhimurium* growth over time.

A comparison of the inhibition halos against *S. enteritidis* and *S. typhimurium* responses reveals analogous patterns. The D condition is characterised by the most protracted bacterial inhibition, followed by the P group, which exhibits a gradual decline in efficacy, and the I group, which is unable to maintain inhibition over an extended period. The reduction in inhibition halos observed across T2 suggests that antimicrobial activity, independent of the condition, may be time-dependent.

Similar comparative dynamics between Gram-negative bacterial strains were observed in the study by Martynowycz et al. [54], indicating that slight differences in outer membrane composition can lead to varying AMP sensitivity.

Molecular tests showed that trends in the expression of cecropins and defensins suggest a differentiated immune response depending on the experimental conditions [35,55,56]. Cecropins, known for their strong antimicrobial activity, exhibited a significant increase in abundance in response to *Salmonella* exposure [37]. In the P group, in which no infection was present, cecropin production remained stable over time, indicating a baseline level of expression in the absence of bacterial challenge. However, in the I group, there was a marked increase in cecropin levels, suggesting that local exposure to the pathogen triggered upregulation of antimicrobial peptides in response. The most pronounced increase was observed in the D group, in which cecropin production nearly tripled, likely reflecting a systemic immune response to infection that required a more robust antimicrobial defense.

Similar upregulation of cecropins in response to bacterial infection was described in the work of Yi et al. [31], wherein they noted strong transcriptional activation upon pathogen exposure.

Defensins, another class of antimicrobial peptides with both bactericidal and immunomodulatory functions, followed a different trend [29]. Unlike cecropins, defensin production in the P group condition declined over time, suggesting that in the absence of infection, the immune system downregulates their expression to maintain immune homeostasis and prevent unnecessary inflammatory responses. Conversely, in the I and D groups, defensins increased over time, indicating their role in responding to bacterial presence. The increase was particularly notable in the D group, suggesting that defensins contribute not only to direct antimicrobial activity but also to the regulation of immune processes in a systemic infection scenario [57].

These patterns are in agreement with observations from Fu et al. [58], who demonstrated that defensins are tightly regulated depending on the inflammatory context and systemic immune response.

The differential expression patterns of cecropins and defensins highlight distinct functional roles in immune defense [31,59,60]. Cecropins appear to be primarily involved in immediate and direct antimicrobial action, rapidly increasing in abundance in response to bacterial presence [61,62]. In contrast, defensins seem to be more tightly regulated, balancing pathogen elimination with the need to control immune activation and inflammation [23,60]. The decline in defensin levels in the P group further supports the idea that these peptides are modulated based on infection status, avoiding excessive immune responses in the absence of pathogens [57].

These observations are consistent with the model proposed by Fu et al. [58], where cecropins act as “first responders” and defensins are involved in fine-tuning the inflammatory environment during prolonged or systemic infections.

Overall, these findings suggest that while cecropins serve as the first line of antimicrobial defense [63], defensins may play a dual role, contributing to both pathogen clearance and immune regulation [64]. The differences in their regulation over time and between experimental conditions underscore the complexity of the immune response and highlight potential mechanisms through which the host fine-tunes antimicrobial peptide production to optimize defense while minimizing unnecessary immune activation [60]. It could also be hypothesized that the two defense pathways are differentially activated.

This fine regulation of AMP expression has been similarly noted by Viljakainen [65], who discussed evolutionary adaptations in AMP regulation to balance immunity and tissue homeostasis in insects.

These findings related to the group subjected only to piercing (P) underscore the critical importance of including, as previously implemented by Bruno et al. [60,66], a control group of this nature to account for background noise arising solely from the physiological response to mechanical stress.

## 5. Conclusions

Our findings demonstrate that antimicrobial peptides (AMPs) produced by *Hermetia illucens* effectively inhibit the growth of both strains of *Salmonella*. Their production is influenced by the type of stress experienced by the larvae, with both biotic and abiotic stress modulating the expression of key peptide families, including defensins and cecropins.

Gene expression analyses and microbiological assays revealed stress-dependent modulation patterns that warrant further investigation. Notably, larvae pricked with a sterile needle (group P) exhibited distinct expression profiles compared to those pricked in the presence of microorganisms (group D). These differences were apparent in both the quantity and the type of the two investigated AMP categories, as well as in their temporal expression patterns. Future studies will focus on isolating the peptide fractions to enable a more precise identification of the peptides involved.

Overall, these results suggest that immune activation in *H. illucens* is subject to complex regulation. Further research is essential to understand the mechanisms that control the immune response, particularly in larvae that have suffered mechanical injury but are not infected with pathogens.

While our findings provide important insights into the antimicrobial peptide response of *H. illucens* larvae under different stress conditions, several limitations should be acknowledged. The relatively small sample size and the focus on discrete time points limit the statistical power and the ability to capture dynamic immune responses over time. Furthermore, the study design did not allow for multifactorial analyses to explore interactions between stress type and temporal factors. Additionally, the use of pooled biological replicates may mask individual variability. These constraints suggest caution when generalizing the results, underscoring the need for future research with expanded sample sizes, longitudinal sampling, and more complex experimental designs to fully elucidate the immune mechanisms involved.

The authors are aware that this study raises more questions than it resolves. Nevertheless, the status that *Hermetia* has attained in recent decades as a farmed insect necessitates intensified research efforts aimed at integrating knowledge across disciplines and research groups. Such a multidisciplinary approach is essential to address the evolving needs of the scientific community, the insect farming industry, and public health authorities. Establishing this foundational knowledge is a prerequisite for more targeted investigations into the physiology of *H. illucens* immune response to specific pathogens. These future studies would not only contribute to a clearer understanding of the insect’s immune architecture but also support the identification of measurable parameters for monitoring the health status of farming systems.

## Figures and Tables

**Figure 1 insects-16-00692-f001:**
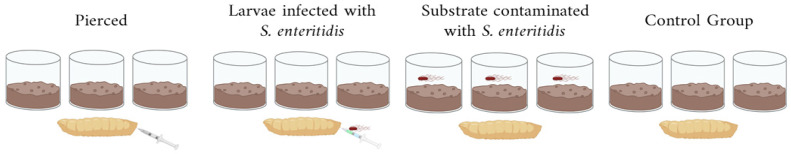
Experimental groups.

**Figure 2 insects-16-00692-f002:**
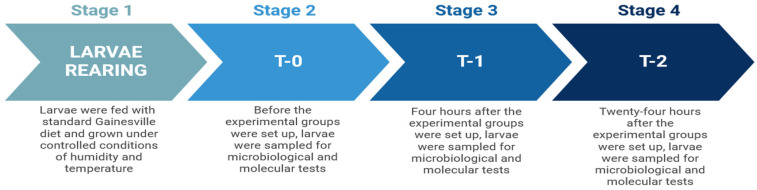
Experimental timeline.

**Figure 3 insects-16-00692-f003:**
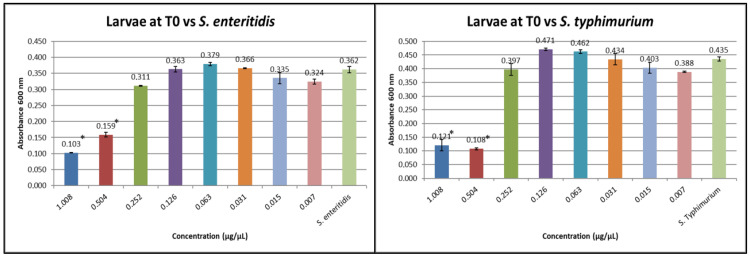
MIC test against *S. enteritidis* and *S. typhimurium* at T0 (the Y axes represent absorbance of the peptide fraction; on the X axes, “concentration” refers to values of serial twofold dilutions of standard solution (from 1.008 µg/µL to 0.007 µg/µL)). (Data are expressed as the means ± standard deviation of three independent biological replicates) (* defined *p* < 0.001 vs. *Salmonella* strain).

**Figure 4 insects-16-00692-f004:**
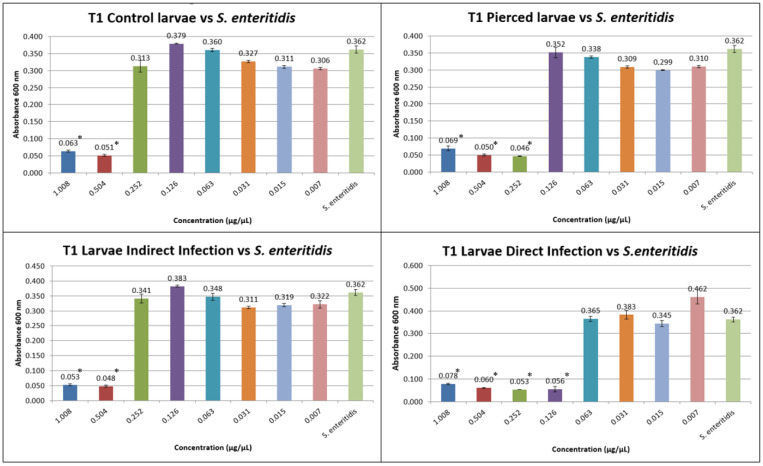
MIC test against *S. enteritidis* at T1 (the Y axes represent absorbance of the peptide fraction; on the X axes, “concentration” refers to values of serial twofold dilutions of standard solution (from 1.008 µg/µL to 0.007 µg/µL)). (Data are expressed as the means ± standard deviation of three independent biological replicates) (* defined *p* < 0.001 vs. *S. enteritidis*).

**Figure 5 insects-16-00692-f005:**
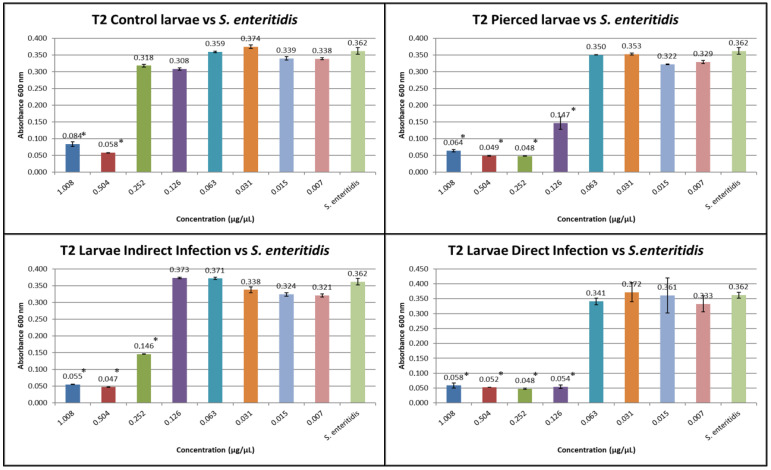
MIC test against *S. enteritidis* at T2 (the Y axes represent absorbance of the peptide fraction; on the X axes, “concentration” refers to values of serial twofold dilutions of standard solution (from 1.008 µg/µL to 0.007 µg/µL)). (Data are expressed as the means ± standard deviation of three independent biological replicates) (* defined *p* < 0.001 vs. *S. enteritidis*).

**Figure 6 insects-16-00692-f006:**
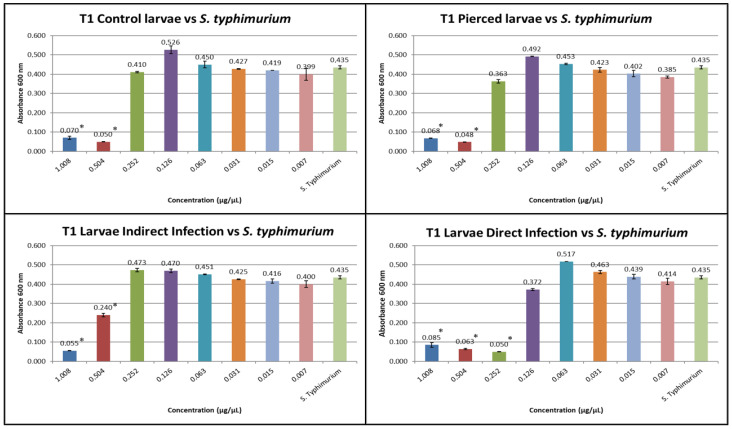
MIC test against *S. typhimurium* at T1 (the Y axes represent absorbance of the peptide fraction; on the X axes, “concentration” refers to values of serial twofold dilutions of standard solution (from 1.008 µg/µL to 0.007 µg/µL)). (Data are expressed as the means ± standard deviation of three independent biological replicates) (* defined *p* < 0.001 vs. *S. typhimurium*).

**Figure 7 insects-16-00692-f007:**
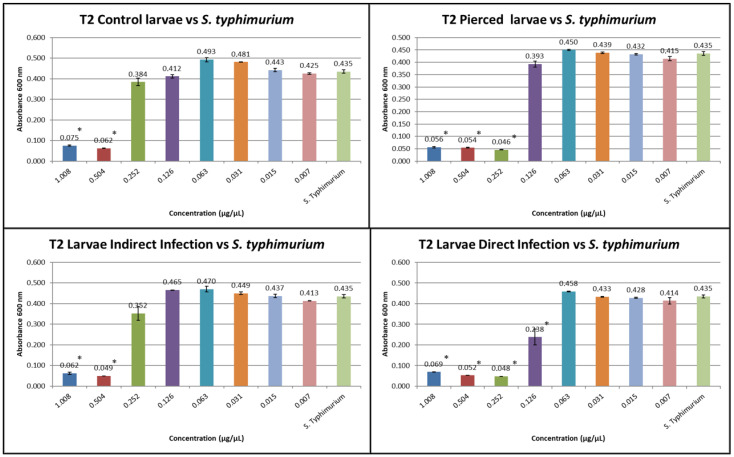
MIC test against *S. typhimurium* at T2 (the Y axes represent absorbance of the peptide fraction; on the X axes, “concentration” refers to values of serial twofold dilutions of standard solution (from 1.008 µg/µL to 0.007 µg/µL)). (Data are expressed as the means ± standard deviation of three independent biological replicates) (* defined *p* < 0.001 vs. *S. typhimurium*).

**Figure 8 insects-16-00692-f008:**
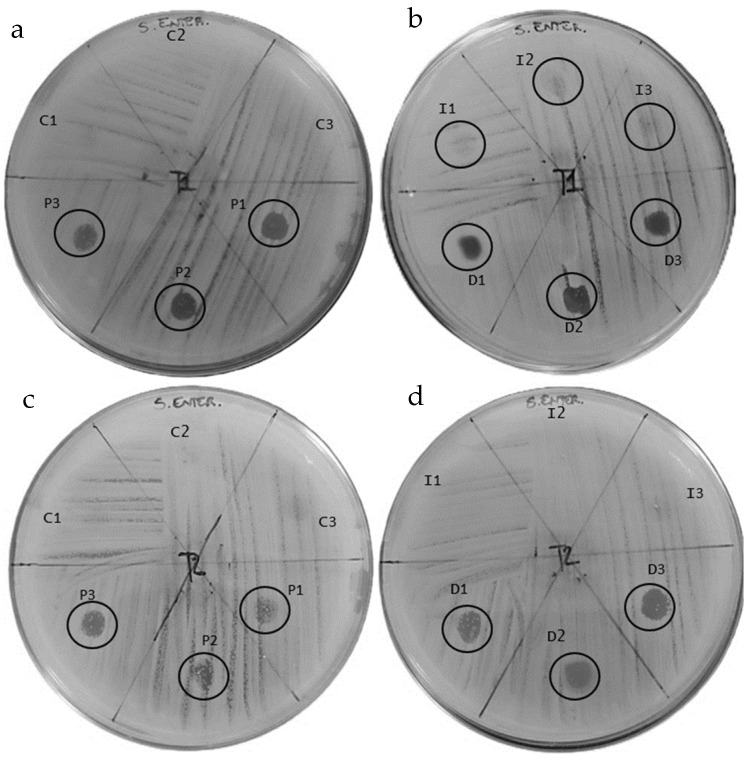
Evaluation of the antibacterial effect of hemolymph against *S. enteritidis* at T1 (**a**,**b**) and T2 (**c**,**d**).

**Figure 9 insects-16-00692-f009:**
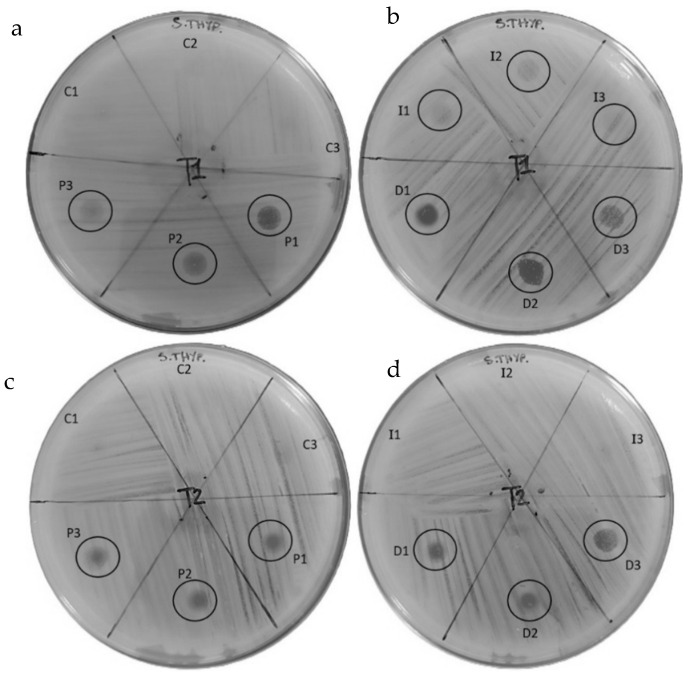
Evaluation of the antibacterial effect of hemolymph against S. typhimurium at T1 (**a**,**b**) and T2 (**c**,**d**).

**Figure 10 insects-16-00692-f010:**
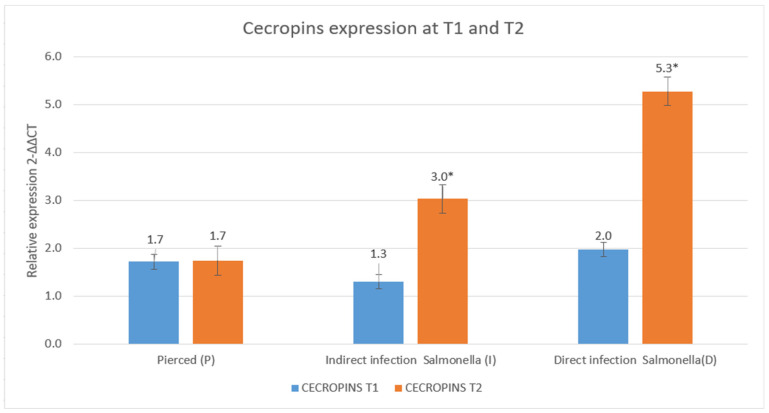
Relative expression of cecropins at T1 and T2 (data are expressed as the means ± standard deviation of three independent biological replicates) (* defined *p* < 0.005 vs. relative expression at T1).

**Figure 11 insects-16-00692-f011:**
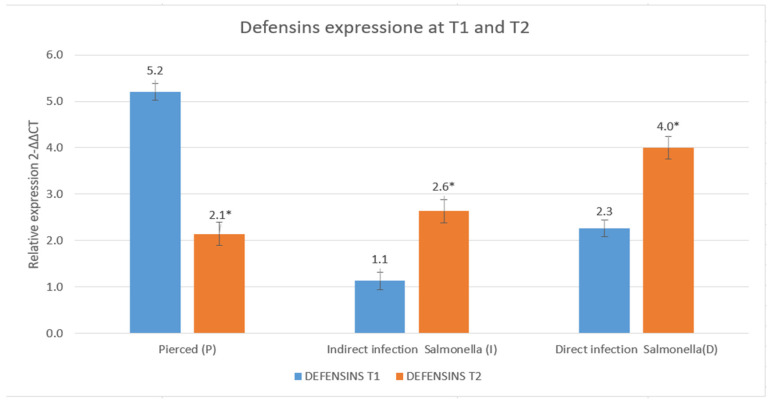
Relative expression of defensins at T1 and T2 (data are expressed as the means ± standard deviation of three independent biological replicates) (* defined *p* < 0.005 vs. relative expression at T1).

**Table 1 insects-16-00692-t001:** Primer sequences of defensins and cecropins.

Name	Sequence 5′–3′
TOT-CECROPINS FW	CGGTCAAAGCGAAGCTGGTT
TOT-CECROPINS RW	TGCCAGAACATTGGCTCCTTG
TOT-DEFENSINS FW	TAGTGGAGCAGCATTATCGGG
TOT-DEFENSINS RW	GTCGTCCCATGGCAATACAATG

**Table 2 insects-16-00692-t002:** Results of antibiogram tests against *S. enteritidis*.

Bacterial Strain: *Salmonella enteritidis*				
Sample	Inhibition Halo (mm)	Colonies	Inhibition Halo (mm)	Colonies
T1			T2	
**C-1**	Absence	Presence	Absence	Presence
**C-2**	Absence	Presence	Absence	Presence
**C-3**	Absence	Presence	Absence	Presence
**P-1**	4.1	Absence	2.1	+++
**P-2**	3.5	+	1.7	+++
**P-3**	3.2	+	2.5	++
**I-1**	2.1	++++	Absence	Presence
**I-2**	2.0	++++	Absence	Presence
**I-3**	1.9	++++	Absence	Presence
**D-1**	3.5	+	4.0	++
**D-2**	4.0	+	3.6	+
**D-3**	3.2	+++	3.4	+

+ = 0 to 5 colonies; ++ = 6 to 10 colonies; +++ = 11 to 15 colonies; ++++ = 16 to 20 colonies.

**Table 3 insects-16-00692-t003:** Results of the antibiogram test against *S. typhimurium*.

Bacterial Strain: *Salmonella typhimurium*				
Sample	Halo of Inhibition (mm)	Colonies	Halo of Inhibition (mm)	Colonies
T1			T2	
**C-1**	Absence	Presence	Absence	Presence
**C-2**	Absence	Presence	Absence	Presence
**C-3**	Absence	Presence	Absence	Presence
**P-1**	4.0	++	2.1	+++
**P-2**	4.1	+++	2.0	+++
**P-3**	3.6	++++	2.7	+++
**I-1**	3.4	Presence	Absence	Presence
**I-2**	3.2	Presence	Absence	Presence
**I-3**	3.1	Presence	Absence	Presence
**D-1**	4.1	+	2.5	++
**D-2**	4.3	+	2.2	+++
**D-3**	3.5	+++	2.7	+++

+ = 0 to 5 colonies; ++ = 6 to 10 colonies; +++ = 11 to 15 colonies; ++++ = 16 to 20 colonies.

## Data Availability

The original contributions presented in this study are included in the article. Further inquiries can be directed to the corresponding author.

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
