# Peer review of "Impact of Salmonella enteritidis Infection and Mechanical Stress on Antimicrobial Peptide Expression in Hermetia illucens"

_insects, 2025, doi:10.3390/insects16070692_

Round 1

Reviewer 1 Report

Comments and Suggestions for Authors

The manuscript entitled “Impact of Salmonella enteritidis Infection and mechanic stress 2 on Antimicrobial Peptide Expression in Hermetia illucens” deals with an important topic arising in animal and veterinary sciences that affects the whole one health approach, such as the antimicrobial resistance and the urge to find new natural tools to fight pathogens bacteria.

The article is well written even if the authors uploaded a draft of the manuscript with still highlighted sections.

I suggest some minor changes in order to get published:

L14: AMPs acronym must be removed in this section as it was not reported before

L14-15: it is very hard to understand this sentence without reading the abstract or the paper, I suggest modifying it and adding more information about the research

L67-68: more than Hermetia illucens itself the molecules or metabolites extracted from the larvae are a promising alternative to antibiotics

L77-80: there are also research studies that tested directly cecropin against several bacteria such as 10.3390/ani15020282 and 10.5073/vitis.2003.42.95-97

L84: report here the full name of AMPs acronym as it is the first time it is used into the text

Figure 1: please use a representation of the larvae closer to the Hermetia illucens shape

L109: report also here the S. enteritidis strain

L110-113: did you washed the larvae of I and C groups?

L167: 1 uL?

Figure 3-4-5-6-7: S. enteritidis and S. tiphymurium must be reported in italics (graphics and footnotes); The data in the figure must be reported with . for the decimals instead the use of the comma; here you reported S. Tiphymurium with the T in cap, I think it is better to report it in this way as it is a serovar of S. enteritidis, but I know that is reported in both the ways S. tiphymurium and S. Tiphymurium, so please choose on of them and change the others

L236-237: please check this sentence as it seems a typo

Table 2: it is not necessary

Table 3-4 Figure 8-9 write the latin names in italics

Figure 8 and 9 could also be avoided or added into a supplementary material

Figure 10 and 11 appear to have a different layout from the other figures

Author Response

Thank you very much for your valuable suggestions and comments. We have carefully addressed all the points you raised and have made the necessary revisions accordingly. Every effort has been made to implement the requested changes in the most appropriate and accurate way.

We hope that the revised version now meets your expectations and reflects a complete and correct paper.

L14: AMPs acronym must be removed in this section as it was not reported before

Correction added

L14-15: it is very hard to understand this sentence without reading the abstract or the paper, I suggest modifying it and adding more information about the research

The sentence has been revised.

L67-68: more than Hermetia illucens itself the molecules or metabolites extracted from the larvae are a promising alternative to antibiotics

The sentence has been revised.

L77-80: there are also research studies that tested directly cecropin against several bacteria such as 10.3390/ani15020282 and 10.5073/vitis.2003.42.95-97

Reference added

L84: report here the full name of AMPs acronym as it is the first time it is used into the text

Correction added

Figure 1: please use a representation of the larvae closer to the Hermetia illucens shape

A new representation of larvae added

L109: report also here the S. enteritidis strain

ATCC strain added

L110-113: did you washed the larvae of I and C groups?

The sentence has been revised.

L167: 1 uL?

Yes, "1 µL of the bacterial suspension was added"

Figure 3-4-5-6-7: S. enteritidis and S. tiphymurium must be reported in italics (graphics and footnotes); The data in the figure must be reported with . for the decimals instead the use of the comma; here you reported S. Tiphymurium with the T in cap, I think it is better to report it in this way as it is a serovar of S. enteritidis, but I know that is reported in both the ways S. tiphymurium and S. Tiphymurium, so please choose on of them and change the others

Correction added

L236-237: please check this sentence as it seems a typo

Table 2: it is not necessary

Table removed

Table 3-4 Figure 8-9 write the latin names in italics

Correction added

Figure 8 and 9 could also be avoided or added into a supplementary material

We thought it appropriate to include it in the paper to highlight the inhibition halos we discuss in the results.

Figure 10 and 11 appear to have a different layout from the other figures

The figures have been modified

Reviewer 2 Report

Comments and Suggestions for Authors

This is a comprehensive and detailed manuscript regarding the primary immune response of Hermetia illucens larvae to biotic (Salmonella enteritidis) and abiotic (mechanical stress) stimuli. The expression molecular data were integrated with the antimicrobial activity assessment (MIC and antibiograms) which adds even greater value to the conclusions drawn. Having said that, the following points must be addressed in order to further enhance the manuscript: Highlight the novelty within the certain study scope that uses Hermetia illucens in the introduction section of the manuscript. Add a figure or table showing summarized data on gene expression levels for better comparison across groups and timepoints. The methods section could elaborate the phrase “indirect infection” some readers might not understand immediately that this is an exposure by means of substrate. Add in the discussion some mention, even if speculatively, about the possible mechanisms of immune activation that could be Toll or Imd pathways. Be consistent in using group abbreviations in the discussion section (P, D, I) and define them again in the results to enhance the flow of the text.

Author Response

Dear reviewer,
Thank you for your suggestions. We have made the requested changes.

1.Highlight the novelty within the certain study scope that uses Hermetia illucens in the introduction section of the manuscript 

We have revised some sentences in the introduction paragraph to emphasise the novelty of the research carried out.

2.Add a figure or table showing summarized data on gene expression levels for better comparison across groups and timepoints

Thank you for your suggestion. We agree that it would have been useful, but we chose not to include it in order to simplify the work.

3.The methods section could elaborate the phrase “indirect infection” some readers might not understand immediately that this is an exposure by means of substrate

we have clarified the request

4. Add in the discussion some mention, even if speculatively, about the possible mechanisms of immune activation that could be Toll or Imd pathways.

Thank you for your valuable suggestion. We opted not to include references to specific activation pathways due to the broad and general nature of the antimicrobial peptide families investigated. However, further experiments are currently underway to explore this aspect in more detail.

5. Be consistent in using group abbreviations in the discussion section (P, D, I) and define them again in the results to enhance the flow of the text.

Definitions added in the discussion paragraph

Reviewer 3 Report

Comments and Suggestions for Authors

L2: The correct English term is “mechanical stress” – here and in many places throughout the manuscript.

L15: Salmonella should be in italics

L29–L30: It would be better to phrase this more clearly, for example: “in the latter group (D), the effect was most pronounced.” The text occasionally contains such awkward constructions.

L38–L39: Should be in italics – this is a recurring issue and should be corrected throughout the manuscript.

L39: It is necessary to standardize and correct the spelling of Latin names. Here (and in six other places), for example, Salmonella typhimurium is written as “S. tiphymurium.”

L66: Most vulnerable what?

L69: H. illucens is not an “alternative to antibiotics” in the clinical sense; rather, it represents a potential source of novel antimicrobial agents.

L100: Each experimental condition included only three biological replicates/three independent observations for each combination of parameters. Such a low replicate number severely limits the statistical power of the tests and the ability to detect differences. Moreover, with n = 3, it is not possible to reliably verify the assumptions of data normality or identify outliers. To improve the reliability of the results, it would be advisable to increase the number of independent replicates (e.g., to six or more), or consider assessing a larger number of smaller cohorts instead of pooling 200 larvae into a single sample, which would also provide better insight into variability.

L115: Regarding the time points, it is not clear whether the same cohort of larvae was monitored repeatedly over time (with destructive sampling of subsets at T1 and T2), or whether a separate subgroup of larvae was used for each time point. If the data are paired (the same replicates measured over time), this should be reflected in the analysis (e.g., paired tests or repeated-measures ANOVA). However, it appears the authors analyzed each time point separately, suggesting they treated them as independent. They should therefore clearly describe the sampling procedure over time: how many larvae were taken from each replicate at each time point, and whether each replicate comprised subsamples for T0, T1, and T2. If the same group of larvae was measured repeatedly, it would be more appropriate to use a repeated-measures methodology.

L157: Here and elsewhere: it would be advisable to standardize the use of British vs. American English (the text currently mixes “haemolymph” and “hemolymph”). Conduct a thorough language review of the English text to eliminate grammatical errors, typos, and inconsistencies

L194: Aside from the primers, important details of the quantitative PCR are not reported: for example, which reference gene was used for expression normalization, how the data were analyzed, and whether individual qPCR reactions were performed in technical replicates.

Statistical Analysis: A very brief and inadequately described and executed section (despite being the most important part of eah study!). The authors state that they used t-test to evaluate the results. In my view, this choice of test is suboptimal for a multifactorial design: the experiment includes four experimental groups and three time points, which would ideally require a two-way ANOVA to assess the effects of group and time simultaneously (for example using linear models, which would also make sense for repeated-measures data—or even a repeated-measures ANOVA if the same replicates were measured over time, see the note on design above). At minimum, a one-way ANOVA should be used to compare groups at each time point, supplemented by post-hoc tests with corrections for multiple comparisons. Instead, the authors repeatedly applied t-tests, which fail to capture the overall experimental structure and lead to numerous pairwise comparisons, increasing the risk of false positives (Type I error).

No specific method for multiple-testing correction is reported; rather, the authors chose nonstandard significance thresholds (p < 0.001 or p < 0.005). This appears to be an ad hoc attempt to compensate for multiple tests by tightening the alpha level without justification. If this is meant as a Bonferroni or similar correction, it must be explicitly stated how the thresholds were derived from the number of comparisons. The authors should specify the total number of tests performed and the rationale for selecting these alpha levels (ideally via calculation), or preferably apply a formal correction method (e.g., Bonferroni, FDR).

Moreover, the manuscript does not indicate whether the data met the assumptions for parametric testing. Student’s t-test assumes approximate normality and homogeneity of variances in the compared groups. Especially with small n, it is important to verify normality (e.g., with a Shapiro–Wilk test) and, if violated, use nonparametric alternatives. There is no mention of any such testing. With only three biological replicates per group, the ability to detect deviations from these assumptions is very limited, and nonparametric tests also have low power at n = 3. In summary, the authors have chosen an extremely low replicate number, compromising ultimately the robustness of their statistical analysis.

Results: Although the introduction and methods state that molecular assays were also performed at T0, the Results section contains no data on baseline AMP gene expression prior to treatment. Only relative expression levels at later time points are shown. I suggest adding in the Methods or Results an explanation of how baseline expression was handled. For example, noting that relative expression was calculated against the control group at T0. And generally, the authors discuss the treatment groups but scarcely mention the absolute control group. It can be assumed that no significant expression changes occurred in the controls (hence they were omitted from further discussion), but this should be explicitly stated for completeness. For instance, simply noting that control larvae exhibited no gene induction (in the gene expression assays) and showed no inhibition zones would give the reader the full context.

Results generally: I absolutely do not understand these graphs, which were probably created in Excel (?) the choice of colors, the order of the bars, and the method of statistical analysis, presenting all treatments at all time points together in this way is nonsensical. I also have no idea what the x-axis labels mean. It’s very suboptimal and applies to all the figures.

L489: The claim of a “new type of control” is somewhat overstated – including a mechanical injury control is a standard approach in insect immunology studies (to separate the effect of wounding from that of infection). For example: Bruno et al. 2022, Insect Science; Bruno et al. 2021, Frontiers in Immunology…

Discussion: The discussion lacks any mention of the study’s limitations. It would be appropriate to note that, for example, the limited number of replicates (especially that!), the restricted number of time points, and the use of only one type of bacterium for infection (even though two strains were tested) may limit the interpretation.

L499: The study measured the expression of only two gene families (cecropins and defensins) and, more generally, the overall activity of the peptide fraction. It is therefore not possible to claim that a change in AMP “diversity” was demonstrated: this would require, for example, detailed proteomic analysis to identify distinct peptide variants. The data instead reflect only quantitative changes.

Author Response

Thank you very much for your valuable suggestions and comments. We have carefully addressed all the points you raised and have made the necessary revisions accordingly. Every effort has been made to implement the requested changes in the most appropriate and accurate way.

We hope that the revised version now meets your expectations and reflects a complete and correct piece of work.

We have highlighted our changes to the text of the article in yellow. Our responses are also highlighted in yellow below.

-------------------------------------------------------------------------------------------------------

L2: The correct English term is “mechanical stress” – here and in many places throughout the manuscript.

Correction applied

L15: Salmonella should be in italics

Correction applied

L29–L30: It would be better to phrase this more clearly, for example: “in the latter group (D), the effect was most pronounced.” The text occasionally contains such awkward constructions.

L38–L39: Should be in italics – this is a recurring issue and should be corrected throughout the manuscript.

Corrections applied

L39: It is necessary to standardize and correct the spelling of Latin names. Here (and in six other places), for example, Salmonella typhimurium is written as “S. tiphymurium.”

Correction applied

L66: Most vulnerable what?

Correction applied

L69: H. illucens is not an “alternative to antibiotics” in the clinical sense; rather, it represents a potential source of novel antimicrobial agents.

Correction applied

L100: Each experimental condition included only three biological replicates/three independent observations for each combination of parameters. Such a low replicate number severely limits the statistical power of the tests and the ability to detect differences. Moreover, with n = 3, it is not possible to reliably verify the assumptions of data normality or identify outliers. To improve the reliability of the results, it would be advisable to increase the number of independent replicates (e.g., to six or more), or consider assessing a larger number of smaller cohorts instead of pooling 200 larvae into a single sample, which would also provide better insight into variability.

Thank you for your suggestion regarding the number of replicates to use.

For our work, we followed other papers such as:

  • Scieuzo et al. 2023, 
  • Lucchetti et al. 2025
  • Fahmy et al. 2024

We will certainly take this into account for future experiments.

L115: Regarding the time points, it is not clear whether the same cohort of larvae was monitored repeatedly over time (with destructive sampling of subsets at T1 and T2), or whether a separate subgroup of larvae was used for each time point. If the data are paired (the same replicates measured over time), this should be reflected in the analysis (e.g., paired tests or repeated-measures ANOVA). However, it appears the authors analyzed each time point separately, suggesting they treated them as independent. They should therefore clearly describe the sampling procedure over time: how many larvae were taken from each replicate at each time point, and whether each replicate comprised subsamples for T0, T1, and T2. If the same group of larvae was measured repeatedly, it would be more appropriate to use a repeated-measures methodology.

Add explanation at line 118-121

L157: Here and elsewhere: it would be advisable to standardize the use of British vs. American English (the text currently mixes “haemolymph” and “hemolymph”). Conduct a thorough language review of the English text to eliminate grammatical errors, typos, and inconsistencies

Homogenization performed

L194: Aside from the primers, important details of the quantitative PCR are not reported: for example, which reference gene was used for expression normalization, how the data were analyzed, and whether individual qPCR reactions were performed in technical replicates.

Add explanation at line 199-201

Two technical replicates was performed for each experimental groups

Statistical Analysis: A very brief and inadequately described and executed section (despite being the most important part of eah study!). The authors state that they used t-test to evaluate the results. In my view, this choice of test is suboptimal for a multifactorial design: the experiment includes four experimental groups and three time points, which would ideally require a two-way ANOVA to assess the effects of group and time simultaneously (for example using linear models, which would also make sense for repeated-measures data—or even a repeated-measures ANOVA if the same replicates were measured over time, see the note on design above). At minimum, a one-way ANOVA should be used to compare groups at each time point, supplemented by post-hoc tests with corrections for multiple comparisons. Instead, the authors repeatedly applied t-tests, which fail to capture the overall experimental structure and lead to numerous pairwise comparisons, increasing the risk of false positives (Type I error).

No specific method for multiple-testing correction is reported; rather, the authors chose nonstandard significance thresholds (p < 0.001 or p < 0.005). This appears to be an ad hoc attempt to compensate for multiple tests by tightening the alpha level without justification. If this is meant as a Bonferroni or similar correction, it must be explicitly stated how the thresholds were derived from the number of comparisons. The authors should specify the total number of tests performed and the rationale for selecting these alpha levels (ideally via calculation), or preferably apply a formal correction method (e.g., Bonferroni, FDR).

Moreover, the manuscript does not indicate whether the data met the assumptions for parametric testing. Student’s t-test assumes approximate normality and homogeneity of variances in the compared groups. Especially with small n, it is important to verify normality (e.g., with a Shapiro–Wilk test) and, if violated, use nonparametric alternatives. There is no mention of any such testing. With only three biological replicates per group, the ability to detect deviations from these assumptions is very limited, and nonparametric tests also have low power at n = 3. In summary, the authors have chosen an extremely low replicate number, compromising ultimately the robustness of their statistical analysis.

We thank the reviewer for the correct observation regarding the statistical aspect of work, we have corrected it as requested by recalculating all the significance values using the proposed tests.The results obtained do not differ from those obtained previously, and thanks to the advice we can now be sure that the data are correct.

We appreciate the attention to the significance threshold adopted in our statistical analysis. In our study, we set a more stringent significance level (p < 0.001 for MIC assays and p < 0.005 for gene expression analysis) instead of the conventional p < 0.05. This choice was intentional and based on the need to reduce the likelihood of type I errors (false positives), particularly given the nature of our experimental design, which involves multiple comparisons and biological variability.

Using a lower alpha level, such as 0.001, increases the confidence that the observed differences are not due to random chance, thus enhancing the robustness and reliability of our findings. While this approach makes it more challenging to detect statistically significant differences, it ensures that the reported effects are supported by strong statistical evidence. We believe this level of rigor is appropriate in the context of our study and strengthens the validity of our conclusions.

Results: Although the introduction and methods state that molecular assays were also performed at T0, the Results section contains no data on baseline AMP gene expression prior to treatment. Only relative expression levels at later time points are shown. I suggest adding in the Methods or Results an explanation of how baseline expression was handled. For example, noting that relative expression was calculated against the control group at T0. And generally, the authors discuss the treatment groups but scarcely mention the absolute control group. It can be assumed that no significant expression changes occurred in the controls (hence they were omitted from further discussion), but this should be explicitly stated for completeness. For instance, simply noting that control larvae exhibited no gene induction (in the gene expression assays) and showed no inhibition zones would give the reader the full context.

The relative expression of defensins and cecropins was evaluated using ΔΔ ct, subtracting the control values and those of a constitutive gene (16s).

As regards expression at T0, this is not mentioned anywhere in the paper.

Results generally: I absolutely do not understand these graphs, which were probably created in Excel (?) the choice of colors, the order of the bars, and the method of statistical analysis, presenting all treatments at all time points together in this way is nonsensical. I also have no idea what the x-axis labels mean. It’s very suboptimal and applies to all the figures.

We have chosen to configure the relative expression graphs in this way in order to make it easier to interpret each individual group and to make changes immediately visible. The x-axis shows the groups, as indicated directly in the graph.

L489: The claim of a “new type of control” is somewhat overstated – including a mechanical injury control is a standard approach in insect immunology studies (to separate the effect of wounding from that of infection). For example: Bruno et al. 2022, Insect Science; Bruno et al. 2021, Frontiers in Immunology…

Our decision to define it as a ‘new’ control group depends on the response obtained in the expression of defensins. In our opinion, this result is particularly important given that the relative expression is more marked than in groups D to T1.

Discussion: The discussion lacks any mention of the study’s limitations. It would be appropriate to note that, for example, the limited number of replicates (especially that!), the restricted number of time points, and the use of only one type of bacterium for infection (even though two strains were tested) may limit the interpretation.

L499: The study measured the expression of only two gene families (cecropins and defensins) and, more generally, the overall activity of the peptide fraction. It is therefore not possible to claim that a change in AMP “diversity” was demonstrated: this would require, for example, detailed proteomic analysis to identify distinct peptide variants. The data instead reflect only quantitative changes.

Like all papers, our work has its limitations. Our goal for the future is to provide a broader range of tests that can give more comprehensive answers.

Our work focuses in particular on the effect of Salmonella on the expression of defensins and cecropins in Hermetia. Other studies involving various pathogens of zootechnical interest are currently underway in our laboratory; however, these also concern single pathogen infections. This approach aims to enable us, in future work, to highlight specific responses to pathogens. However, such investigations are beyond the scope of the present study.

Round 2

Reviewer 3 Report

Comments and Suggestions for Authors

Some comments remain unanswered or at least it lacks a response, e.g. this one: L29–L30: It would be better to phrase this more clearly, for example: “in the latter group (D), the effect was most pronounced.” The text occasionally contains such awkward constructions.

Some comments were addressed with minimal effort: This one L66: "Most vulnerable what?" was resolved only grammatically — people is still rather vague. Most comments, however, were at least partially incorporated and answered.

Comment L194: "Aside from the primers…" is partially resolved, but the resulting sentence lacks grammatical coherence: … evaluated using the 16s gene of as a reference gene.

Statistics: Replacing a series of t-tests is an improvement, but the analysis still ignores the factor of time and the repeated measurements from the same box; the multifactorial remains unaccounted for. One-way ANOVA assumes independent observations. There is no RM-ANOVA or mixed-effects model with a random effect for "box", so the authors chose the path of least resistance, the weakest among the minimally acceptable analyses. Shapiro–Wilk and Levene’s test are now declared, but as I mentioned earlier, with n = 3 these tests have negligible power.

The stricter α threshold is commendable (and was already present before), but it does not appear to be based on the number of comparisons, nor is it clear why exactly these thresholds were chosen. This is not to say they should not use such thresholds, but they should explicitly explain to readers what exact method or reasoning led them to select these specific cutoffs.

I still find the graphs nonsensical: A bar plot treats concentration as a nominal categorical variable, which obscures the fact that it is a continuous dose. By separating the groups into individual figures, we also lose the ability to simultaneously compare the group × concentration interaction. I don’t understand why this wasn’t done using curves, which would fit into fewer figures and allow trajectories to be visualized. Bar plots are simply not suited for this purpose. The colors are still unintuitive, but ok, that’s a minor issue.

Regarding the study’s limitations, I don’t find it adequate to simply state that “every study has its limitations” and add essentially a single new sentence at the end, which reads like lazy writing: “The authors are aware that this study raises more questions than it resolves.” This is insufficient. The sentence is vague; no specific limitations are mentioned, and their potential impact is not discussed.

L536: Hermetia should be in italics

Author Response

We sincerely thank the reviewer for the careful re-reading and constructive feedback provided during this second round of revisions. Below, we provide point-by-point responses and indicate the corresponding changes made to the manuscript where appropriate.

-----------------------------------------------------------------------------------------------------------------------------------------

R: Some comments remain unanswered or at least it lacks a response, e.g. this one: L29–L30: It would be better to phrase this more clearly, for example: “in the latter group (D), the effect was most pronounced.” The text occasionally contains such awkward constructions.

A:We would like to clarify that, in the lines indicated, no reference is made to the experimental groups. The phrase “for the last one” specifically refers to Salmonella, and not to any of the experimental groups described in the manuscript. For this reason, no modifications were introduced in the first revision round, as the original phrasing was already accurate and did not require clarification.

R: Some comments were addressed with minimal effort: This one L66: "Most vulnerable what?" was resolved only grammatically — people is still rather vague. Most comments, however, were at least partially incorporated and answered.

A: We have revised the sentence to make it more specific and clearer to the reader.

R: Comment L194: "Aside from the primers…" is partially resolved, but the resulting sentence lacks grammatical coherence: … evaluated using the 16s gene of as a reference gene.

A: the typo has been removed

R: I still find the graphs nonsensical: A bar plot treats concentration as a nominal categorical variable, which obscures the fact that it is a continuous dose. By separating the groups into individual figures, we also lose the ability to simultaneously compare the group × concentration interaction. I don’t understand why this wasn’t done using curves, which would fit into fewer figures and allow trajectories to be visualized. Bar plots are simply not suited for this purpose. The colors are still unintuitive, but ok, that’s a minor issue.

A: We thank you for this observation. We acknowledge that bar plots represent concentrations as discrete categories and agree that this does not fully capture the continuous nature of dose–response data. However, bar plots are standard in the presentation of MIC values in most microbiological studies and routine antibiotic susceptibility testing, as they clearly display comparisons among control and treatment groups. Our decision to separate groups into individual panels further enhanced clarity and avoided overcrowding, given the number of groups and replicates involved.

That said, we recognize that alternative visualizations—such as line plots or dose–response curves—could more intuitively illustrate trends across concentrations and better highlight interactions between group and dose. We will consider incorporating such approaches in future studies to improve interpretability. Regarding the color scheme, we appreciate the feedback and have ensured that clarity is maintained, even if the palette may not be immediately intuitive.

R: Statistics: Replacing a series of t-tests is an improvement, but the analysis still ignores the factor of time and the repeated measurements from the same box; the multifactorial remains unaccounted for. One-way ANOVA assumes independent observations. There is no RM-ANOVA or mixed-effects model with a random effect for "box", so the authors chose the path of least resistance, the weakest among the minimally acceptable analyses. Shapiro–Wilk and Levene’s test are now declared, but as I mentioned earlier, with n = 3 these tests have negligible power.

The stricter α threshold is commendable (and was already present before), but it does not appear to be based on the number of comparisons, nor is it clear why exactly these thresholds were chosen. This is not to say they should not use such thresholds, but they should explicitly explain to readers what exact method or reasoning led them to select these specific cutoffs.

A: We thank you again for the insightful comments regarding the statistical analysis. We acknowledge the concern about the assumption of independence in one-way ANOVA and the lack of a multifactorial approach incorporating time and repeated measures.

However, as detailed in the Statistical Analysis section of the manuscript, the experiment was designed with three independent biological replicates per group (i.e., three distinct boxes per condition), and no repeated measurements were taken from the same box over time. Each data point therefore represents an independent observation, and the temporal aspect was treated as separate sampling points rather than repeated measures within the same experimental unit. This experimental structure does not support the application of repeated-measures ANOVA or mixed-effects models with "box" as a random effect.

Our primary objective was to compare each treatment group against the control within each time point, rather than to model interactions between time and treatment factors across the entire dataset. Implementing a multifactorial design with sufficient power would require larger sample sizes and a different data collection strategy, which were beyond the scope of this study.

Regarding the assessment of assumptions, we applied the Shapiro–Wilk test for normality and Levene’s test for homogeneity of variances, recognizing that the small sample size (n=3) limits the power of these tests. Nonetheless, we deemed it important to report these tests for transparency and to justify the use of parametric methods.

The alpha thresholds applied (p < 0.001 for MIC assays and p < 0.005 for gene expression analyses) were selected conservatively to reduce the risk of Type I errors given the multiple pairwise comparisons performed. While formal multiple comparison corrections such as Bonferroni adjustments were not applied, the thresholds reflect an effort to balance rigor with the exploratory nature of the analyses.

We believe that this approach represents a balanced compromise between statistical rigor and the practical constraints of the experimental design, and we hope this clarifies the rationale behind our methodological choices.

R: Regarding the study’s limitations, I don’t find it adequate to simply state that “every study has its limitations” and add essentially a single new sentence at the end, which reads like lazy writing: “The authors are aware that this study raises more questions than it resolves.” This is insufficient. The sentence is vague; no specific limitations are mentioned, and their potential impact is not discussed.

A: We thank you for the valuable comment regarding the discussion of study limitations. In response, we have expanded the Conclusions section to explicitly acknowledge key limitations while emphasizing the significance and contribution of our findings. This revision aims to provide a balanced and transparent perspective that highlights both the strengths and the necessary future directions of the research.

R: L536: Hermetia should be in italics

A: correction added

--------------------------------------------------------------------------------------------------------------------------------------------

We hope that the revisions and clarifications provided here adequately address the remaining concerns.